# Increased Expression of Hepatic Stearoyl-CoA Desaturase (SCD)-1 and Depletion of Eicosapentaenoic Acid (EPA) Content following Cytotoxic Cancer Therapy Are Reversed by Dietary Fish Oil

**DOI:** 10.3390/ijms24043547

**Published:** 2023-02-10

**Authors:** Md Monirujjaman, Leila Baghersad Renani, Peter Isesele, Abha R. Dunichand-Hoedl, Vera C. Mazurak

**Affiliations:** Division of Human Nutrition, Department of Agricultural Food and Nutritional Science, Li Ka Shing Centre for Health Research Innovation, University of Alberta, Edmonton, AB T6G 2P5, Canada

**Keywords:** chemotherapy, colorectal cancer, CASSH, fatty liver, irinotecan, n-3 PUFA, tumor, 5-FU

## Abstract

Cancer treatment evokes impediments to liver metabolism that culminate in fatty liver. This study determined hepatic fatty acid composition and expression of genes and mediators involved in lipid metabolism following chemotherapy treatment. Female rats bearing the Ward colon tumor were administered Irinotecan (CPT-11) +5-fluorouracil (5-FU) and maintained on a control diet or a diet containing eicosapentaenoic acid (EPA) + docosahexaenoic acid (DHA) (2.3 g/100 g fish oil). Healthy animals provided with a control diet served as a reference group. Livers were collected one week after chemotherapy. Triacylglycerol (TG), phospholipid (PL), ten lipid metabolism genes, leptin, and IL-4 were measured. Chemotherapy increased TG content and reduced EPA content in the liver. Expression of SCD1 was upregulated by chemotherapy, while dietary fish oil downregulated its expression. Dietary fish oil down-regulated expression of the fatty acid synthesis gene FASN, while restoring the long chain fatty acid converting genes FADS2 and ELOVL2, and genes involved in mitochondrial β-oxidation (CPT1α) and lipid transport (MTTP1), to values similar to reference animals. Neither leptin nor IL-4 were affected by chemotherapy or diet. Depletion of EPA is associated with pathways evoking enhanced TG accumulation in the liver. Restoring EPA through diet may pose a dietary strategy to attenuate chemotherapy-associated impediments in liver fatty acid metabolism.

## 1. Introduction

Colorectal cancer (CRC) is one of the top three causes of cancer death in North America [1]. CRC most commonly metastasizes to the liver, which is treated surgically in ~15–20% of cases for increased long-term survival [2]. Four or more cycles of chemotherapy are commonly provided before liver resection to reduce tumor size, increase curative resection rates, and to convert some patients from having unresectable to resectable disease [3,4]. Despite improved efficacy and outcomes, commonly used chemotherapeutic drugs such as Irinotecan (CPT-11) and 5-fluorouracil (5-FU) are associated with hepatic injury characterized as steatosis and steatohepatitis, often referred to as chemotherapy-associated steatosis/steatohepatitis (CASSH) [5]. Steatosis is characterized by abnormal lipid accumulation in hepatocytes, which may lead to steatohepatitis accompanied by inflammatory changes and ballooning degeneration in the liver [6,7]. Steatohepatitis can lead to fibrosis, cirrhosis, impaired liver function, and ultimately liver failure in some patients [5,8,9]. Several mechanisms are proposed for the development of steatosis, which include increased de novo lipogenesis within the hepatocytes, increased uptake of fatty acids (from peripheral tissue), reduced utilization through β-oxidation, and reduced secretion of very-low-density-lipoproteins, alone, or in combination with a concomitant increase of oxidative stress [7,10,11]. Steatosis is problematic in oncology settings because it is associated with an increased risk of post-operative complications and mortality following hepatic surgery [6,12].

The liver plays a central role in fatty acid metabolism. Cancer treatment evokes impediments to liver function [13]. It is of interest that depletion of phospholipids (PLs) and triglycerides (TGs) in the plasma of advanced cancer patients and those receiving chemotherapy treatments of various types have been observed [14]. In these patients, shorter survival was associated with depletion of essential fatty acids in blood and muscle [14,15,16]. Patients with advanced cancer (of various tumor types) exhibit a deficit of 30–50% of plasma PL and constituent fatty acids compared with healthy subjects [17], which might be exacerbated by chemotherapy treatment. It was recently revealed that the hepatic content of n-6 and n-3 polyunsaturated fatty acids (PUFAs) is reduced, and genes involved in pathways of PUFA synthesis are down-regulated, by chemotherapy treatments [18]. Collectively, these observations suggest impediments in lipid metabolism in the liver that could potentially impact peripheral availability of essential fatty acids.

Leptin is an adipocyte-secreted adipokine that plays a role in the regulation of energy homeostasis [19]. Leptin has been shown to have an anti-steatotic effect [19] while acting as a pro-fibrogenic and tumorigenic factor [20]. Interleukin (IL)-4 plays a role in the energy metabolism and pathogenesis of obesity [21] and has been shown to increase hepatic TG content by facilitating free fatty acid uptake and expression/activity of lipogenic enzymes in experimental models [21] (Figure 1 shows the effect of leptin and IL-4 on hepatic lipid metabolism). Whether hepatic levels of leptin and IL-4 are altered by the presence of a tumor and/or chemotherapy treatment is unknown.

The beneficial effects of n-3 PUFAs, particularly eicosapentaenoic acid (EPA) and docosahexaenoic acid (DHA), in dyslipidemia and obesity are well documented [22,23]. Studies suggest that dietary fish oil may improve fatty liver [24,25]. Other studies have also shown that dietary n-3 PUFAs can decrease intrahepatic TG content and steatohepatitis [26,27,28]. The lipid-lowering effects of dietary fish oil in the liver are associated with modulation of the expression of key genes involved in liver lipid metabolism [25,29]. Since chemotherapy treatment alters lipid metabolism, as well as the hepatic content of n-6 and n-3 PUFAs and the expression of genes involved in pathways of PUFA synthesis [18], the aim of the present study was to determine whether provision of dietary fish oil containing EPA + DHA initiated at the same time as chemotherapy treatment mitigates altered hepatic lipid content and composition and genes involved in lipid metabolism. The effect of dietary fish oil on leptin and IL-4 in the liver was also investigated.

## 2. Results

### 2.1. General Findings

Relative food intake declined by ~60% after chemotherapy treatment, which was restored to baseline levels by Day 4 (average feed intake 10 ± 0.7 gm per day, which provided ~230 mg fish oil to animals fed on fish oil diet) and remained similar between groups thereafter. Body weight, liver weight, and average liver weight as a percent of body weight (141–157 gm, 5.3–5.7 gm, and 3.5–3.8%, respectively) was not different between groups.

### 2.2. Liver Triglyceride (TG) and Phospholipid (PL) Fatty Acids

TG content in the liver was not significantly different between the Reference and Tumor groups (Table 1(a)); however, the Tumor group had a small but significantly higher amount of palmitic acid (C16:0) and total saturated fatty acid (SFA) content compared to the Reference group. Chemotherapy treatment significantly increased total TG content (2-fold) in the Chemo group compared to the Tumor group, concurrent with a 30% reduction in EPA content. Dietary fish oil had no effect on total TG content; however, lower oleic acid (C18:1n-9) accompanied the increase in EPA (C20:5n-3) and DHA (C22:6n-3) in the Chemo + Fish Oil group (Table 1(a)).

Total liver PL was not different between the Tumor and Reference groups. The Tumor group had lower arachidonic acid (ARA) content and higher EPA content compared to the Reference group (Table 1(b)). Total PL fatty acid was significantly higher (29%), with a lower EPA content (28%), in the Chemo group compared to the Tumor group. The provision of dietary fish oil significantly reduced ARA content with a concurrent increase of EPA and DHA content (Table 1(b)) in liver PL.

### 2.3. Gene Expression

Ten genes involved in the synthesis, oxidation, transport, and overall regulation of lipid metabolism were analyzed (Figure 2). Relative expression of delta-6 desaturase (FADS2) and AMP-activated protein kinase (AMPK) were lower in the Tumor group compared to the Reference group. Chemotherapy treatment increased the expression of stearoyl-CoA desaturase (SCD1) and peroxisome proliferator-activated receptor gamma (PPARγ) in the liver when compared with animals bearing tumors only. The Chemo + Fish Oil group had lower relative expression of fatty acid synthase (FASN) and SCD1, with increased expression of FADS2, fatty acid elongases 2 (ELOVL2), microsomal triglyceride transfer protein-1 (MTTP1), and carnitine palmitoyl transferase 1 (CPT1α) compared to the Chemo group. Provision of dietary fish oil normalized the expression of these genes to values not different from the Reference group (Figure 2). Expression of sterol retinol ester binding protein-1 (SREBP1) and forkhead box O1 (FOXO1) were not affected by tumors, chemotherapy, or dietary fish oil.

### 2.4. Hepatic Leptin and Interleukin (IL)-4

Hepatic leptin was higher, while IL-4 was lower in the Tumor group compared to all other groups (Figure 3). Chemotherapy treatment or chemotherapy + dietary fish oil had no effect on them (Figure 3).

## 3. Discussion

This study reveals the elevation of total hepatic fatty acids both in TG and PL fraction with concurrent reduction of EPA after provision of chemotherapy agents that commonly cause hepatic steatosis and/or steatohepatitis. Dietary fish oil significantly increased EPA and DHA in liver TG and PL. This change in essential fatty acid balance occurred concurrently with a decrease in the key rate-limiting genes SCD1 and FASN, involved in the conversion, and MTTP1 and CPT1α, involved in the transport of fatty acids in the liver, respectively. Down-regulation of genes involved in saturated and monounsaturated fatty acid synthesis was observed. Alterations in lipid metabolism in cancer patients treated with chemotherapy are poorly understood. The single previous report on liver fatty acid composition following chemotherapy treatment was limited to high dose CPT-11 only [30], and recently we have shown alteration of liver fatty acid content and composition aligned with the alteration of expression of genes involved in PUFA conversion [18]. However, to date, no study has determined the effect of dietary fish oil on specific hepatic perturbations in fatty acids following chemotherapy and what changes in genes may underly these changes.

Hepatic fat accumulation, as a consequence of cancer chemotherapy, especially for colorectal cancer, is poorly understood. Hepatic fat accumulation, occurring in obesity and metabolic syndrome in patients with non-alcoholic fatty liver disease (NAFLD) is more well understood [31,32]. In both clinical and preclinical studies, NAFLD is associated with significant changes in the type and abundance of hepatic lipids, characterized by elevation of SFA and MUFA current with lower content of n-3 and n-6 PUFAs [20,21,22,23,24]. In preclinical models of NAFLD, hepatic n-3 PUFAs were more severely depleted than n-6 PUFA [21,25,26]. Similarly, here we show that hepatic EPA content is depleted by chemotherapy treatment. In humans with cancer, evaluation of plasma PL reveals a depletion of essential PUFAs in cancer patients at diagnosis, which is exacerbated during cancer chemotherapy treatments [9,11,27]. Depletion of essential PUFAs in the liver following chemotherapy treatment could be one reason why low levels of essential fatty acids have been observed in the plasma of patients being treated for cancer [9].

The effects of dietary n-3 PUFAs in reducing the accumulation of liver lipids have been shown in several experimental animals, including NAFLD and metabolic syndrome [28,33,34,35]. However, results from clinical studies evaluating the effect of n-3 PUFAs in reducing hepatic fat content in NAFLD are contradictory [36,37,38,39], and no study has evaluated the effect of dietary n-3 PUFA on hepatic lipid metabolism in cancer animal models provided with chemotherapy treatments. In the present study, we show that hepatic MUFA contained within the TG fraction is significantly reduced while n-3 PUFA content increased in a cancer animal model provided dietary fish oil immediately following the provision of chemotherapy. Emerging studies have demonstrated that fish oil containing EPA+DHA can ameliorate mitochondrial dysfunction [40], stimulate fatty acid β-oxidation [41], inhibit lipogenesis [41,42], and restore lipid homeostasis at the adipose tissue–liver axis [43] in the liver.

Chemotherapy treatment significantly upregulated the expression of SCD1 and PPARγ, while dietary fish oil containing EPA + DHA reduced the expression of SCD1 and FASN to levels similar to reference animals. Our findings are similar to previous findings in which endogenous n-3 PUFAs significantly decreased the expression of SCD1 and/or FASN either at the transcriptional or protein levels [44,45]. In the liver, SCD1 is tightly regulated in the biosynthesis of palmitoleate (C16:1 n-7) and oleate (C18:1 n-9), major components of TG, from palmitic acid (C16:0) and oleic acid (C18:0) [42,44], while PPARγ promotes free fatty acid uptake and enhance lipogenesis by inducing specific genes [46,47,48]. Both SCD1 and PPARγ facilitate an increase in hepatic TG [47,48,49,50,51]. Higher hepatic expressions of SCD1 [52,53] and PPARγ [54] have been observed in NAFLD patients. SCD1 is modified by dietary components [55], which aligns with our study, where dietary fish oil reduced the expression of SCD1 and was reflected in the direct reduction of oleic acid and total MUFA content in the liver (Table 1(a)). Although dietary fish oil did not reduce PPARγ expression in our model, reduction of FASN, the downstream target of PPARγ, would be expected to reduce hepatic de novo lipogenesis.

The FADS2 gene that encodes the delta-6 desaturase enzyme is one of two rate-limiting enzymes that convert the PUFA precursors linoleic acid (C18:2n-6) and alpha-linolenic acid (C18:3n-3) to their respective metabolites, ARA and EPA/docosapentaenoic acid (DPA, C22:5n-3), respectively [28,29,30]. The ELOVL2 gene encodes the enzyme that regulates the conversion of ARA and EPA to adrenic acid (C22:4n-6) and DPA, respectively, as well as the conversion of DPA to tetracosapentaenoic acid (C24:5n-3) [31]. Although both FADS2 and ELOVL2 genes appeared to be unaltered by chemotherapy, up-regulation of their expression by dietary fish oil suggests that pathways regulating conversion of PUFA to longer unsaturated products, are positively impacted by dietary fish oil containing EPA+DHA, which was reflected in the individual as well as total PUFA contents in the liver in our animals (Table 1(a,b)). Since chemotherapy resulted in the depletion of hepatic n-3 PUFA content [18], increasing n-3 PUFA content might be protective against essential PUFA depletions in cancer patients who are undergoing chemotherapy treatments.

CPT1α is a regulatory enzyme that transfers fatty acids from the cytosol to the mitochondria prior to β-oxidation [56]. Inhibition of CPT1α leads to mitochondrial dysfunction [57,58], and its expression in humans is reduced by 50% in NAFLD in the liver [59]. Dietary fish oil restored CPT1α gene expression and expression of AMPK to levels similar to reference animals. AMPK maintains energy homeostasis and is involved in multiple aspects of anti-lipid metabolism, and can increase fatty acid oxidation by upregulating expression of CPT1α [60]. Activation of the AMPK-CPT1α signaling pathway reduced lipid deposition and improved lipid metabolism in NAFLD [61] and in other conditions [62,63]. Similarly, CPT1α gene therapy reduced diet-induced hepatic steatosis in mice [64]. Our study finding is consistent with a previous rat study in which dietary EPA + DHA led to a 50% increase of activity of the CPT1α enzyme with concurrent increase of rate of mitochondrial β-oxidation in the liver [65]. It has been shown that increased CPT1α activity attenuates hepatic steatosis [66]. Although we did not measure the CPT1α activity, it is speculated that higher expression also leads to higher activity; however, this needs to be examined in future studies.

The liver packages fatty acids from the diet and those synthesized in the body into very low-density lipoproteins (VLDL), which are secreted into circulation for distribution to the peripheral tissues, including skeletal muscle, cardiac muscle, and adipose tissue. TGs are packaged together with ApoB-100 into VLDL in the endoplasmic reticulum by the activity of MTTP and secreted into circulation. Hepatic steatosis has been reported in subjects with mutations in MTTP [64], and pharmacologic inhibition or genetic deletion of MTTP caused hepatic steatosis [67,68]. In this study, we have shown that dietary fish oil significantly increased hepatic MTTP1 expression associated with reduction of liver lipid accumulation, and pharmacological upregulation of this protein has been proposed to be a potential treatment option for NAFLD [69].

Leptin is involved in lipid metabolism and, under healthy conditions, suppresses hepatic lipogenesis [70,71,72]. The anti-steatosis action of leptin has been shown in non-obese mice with type I diabetes, whereby leptin administration led to significant reductions of lipogenic transcription factors and decreased plasma and tissue lipids [73]. Similarly, leptin administration improved or prevented hepatic steatosis development in ob/ob mice [74]. IL-4 has been shown to increase TG content in the liver by facilitating free fatty acid uptake and expression/activity of lipogenic enzymes [21]. In our study, leptin level was higher while IL-4 level was lower in the tumor animals compared to other animals. These alterations are parallel with the lower, albeit insignificant, total TG in the liver of the tumor animals. Whether altered leptin and/or IL-4 in tumor animals had any effect on hepatic TG and PL alterations needs to be verified in future studies.

The amount of fish oil used in the current diet is higher than what would be achievable in analogous human diets. The amount of fish oil consumed by each rat was 230 mg/d, which is equivalent to 14.9 g fish oil per day for a 60 kg adult person, which is difficult to achieve in a normal condition. However, the diets used herein were similar to those used to reduce tumor growth in [15], and had a similar amount of EPA + DHA used in clinical studies [75,76].

There are several limitations of our study. First, our study is relatively short as only one cycle of chemotherapy was provided, whereas four or more cycles of chemotherapy are commonly provided before liver resection in the clinical setting. Secondly, changes in gene expression do not necessarily align with protein expression and function, which provides a fertile ground for future investigation. Thirdly, results obtained from animal models cannot simply be extrapolated to humans, since lipid metabolism differs between species. However, the experimental model used for this study carefully recapitulates therapy for colorectal cancer in humans with respect to doses and toxicity of a combined regimen of CPT-11 + 5-FU [77,78]. Moreover, we observed a strong dietary effect even after a short time of dietary intervention initiated at the same time as chemotherapy treatment, during a time when there was a marked reduction in food intake. In our study, we used female animals, since this disease model was developed in female animals only. Lipid metabolism and distribution are affected by sex [79]; therefore, the results described in the present study may not be (completely) transferable to male animals. It would also be important to understand how alterations in essential fatty acids in the liver interplay with peripheral tissues, such as adipose tissue and muscle, and the impact of these changes on whole-body metabolic processes.

## 4. Materials and Methods

### 4.1. Animal Handling, Diet, and Experimental Design

Animal use was reviewed and approved by the University of Alberta Animal Care Committee and conducted in accordance with the Guidelines of the Canadian Council on Animal Care. Female Fischer 344 rats (11–12 weeks older with body weight 109–145 g) were obtained from Charles River (QC, Canada) and were housed two per cage in a room with controlled temperature (22 °C) and 12 h light/dark cycles. Water and food were provided ad libitum under aseptic conditions that included a positive-air-pressured room and filter-top cages. Rats were acclimated for one week prior to the start of the experiment, then fed a semi-purified diet based on the American Institute of Nutrition (AIN)-76 basal diet for laboratory rodents (Harlan Teklad, Indianapolis, IN, USA) with a modified fat component similar to North American dietary pattern (40% of energy, polyunsaturated: saturated fat ratio of 0.35) [15,18]. The experimental fish oil diet contained the same proportion of macronutrients as the control diet, differing only in the addition of 2.3 g fish oil/100 g diet (Ocean Nutrition Canada, Dartmouth, NS, Canada). Added fish oil replaced 2.3 g of other fat in the diet, such that the total fat content (20 g/100 g diet) and the polyunsaturated to saturated fat ratio were similar between control and fish oil diets, as shown in Table 2 and detailed in previous studies of the same animal model [80]. Enrichment of the muscle occurred within 7 days at physiologically relevant levels [81]. This enrichment thus represents a biologically relevant dose of n-3 with a resulting n-6: n-3 fatty acid ratio of 3:2, which is identical to the n-6: n-3 ratio reported for humans in muscle phospholipid after 3 weeks of supplementation with fish oil (2.4 g per day EPA + DHA) [82]. Diet was freshly prepared biweekly and stored in sealed containers at −20 °C until feeding. Routine examination of texture, odor, and color indicated that the oils were not oxidized.

After two weeks of basal diet, rats (*n* = 18) underwent tumor implantation as previously described [15]. Tumor volume was estimated as (cm^3^) = 0.5 × L × W × H using calipers [75]. Rats were housed individually one week prior to receiving chemotherapy treatment. When tumor size reached 2 cm^3^, rats (*n* = 6) in the tumor group were euthanized, while other rats (*n* = 12) were treated with chemotherapy (CPT-11 plus 5-FU). Atropine (1 mg/kg body weight, subcutaneous) was administered to alleviate early onset of cholinergic symptoms immediately prior to CPT-11 (50 mg/kg body weight, intraperitoneal) and 5-FU injection (50 mg/kg body weight, intraperitoneal, administrated one day after CPT-11 administration). This experimental model carefully recapitulates therapy for colorectal cancer in humans with respect to doses and toxicity of a combined regimen of CPT-11 plus 5-FU [77,78,80]. One group (*n* = 6) of tumor-bearing rats receiving chemotherapy treatment were randomly assigned to the fish oil diet (designated as Chemo + Fish Oil group throughout the manuscript) on the same day as CPT-11 was given, while another group (*n* = 6) of tumor-bearing animals receiving chemotherapy continued the control diet (designated as Chemotherapy or Chemo group throughout the manuscript). Tumor-bearing animals not receiving the chemotherapy treatment (*n* = 6), designated as Tumor group, were provided with the control diet. Healthy rats (*n* = 6) did not undergo tumor implantation nor receive chemotherapy, consumed only the control diet (designated as Reference group throughout the manuscript), and were otherwise handled in the same manner as the experimental groups. Figure 4 shows the study design, and fatty acid composition in the diet is presented in Table 2.

Food intake was measured every other day prior to the initiation of chemotherapy, and every day after chemotherapy was administered. Body weight was recorded on the same days as tumor volume. At termination, body weight was converted to tumor-free body weight for data interpretation and statistical analysis. All rats were euthanized with carbon-dioxide asphyxiation. At termination, the livers were collected, weighed, and immediately snap-frozen in liquid nitrogen and stored at −80 °C until further analysis.

### 4.2. Fatty Acid Analysis

Liver TG and PL content and composition were evaluated as previously described [15,16,18]. Briefly, rat liver (50 mg) was homogenized with calcium chloride solution (0.025%) using a sonicator to obtain a uniform mixture. A modified Folch method was used for extraction of total lipids from liver [83]. The PL and TG fraction were isolated using thin layer chromatography. Bands were visualized, scraped, and C15:0 and C17:0 (0.05 µg; Supelco, Bellefonte, PA, USA; Sigma Chemical, St. Louis, MO, USA) added to enable quantification of fatty acids in TG and PL fractions, respectively. Fatty acid methyl esters were determined using gas chromatography (Varian 3600CX Gas Chromatograph, Palo Alto, CA, USA) equipped with a flame ionization detector and BP-20 fused capillary column (SGE Instruments, Victoria, Australia). Fatty acid content was calculated as proportions (%) as well as absolute amounts (μg/g) based on commercially available standards containing a known fatty acid composition.

### 4.3. RNA Preparation and Gene Expression

Total RNA was extracted from liver samples (30 mg) using the RNeasy Mini Kit (Qiagen, Toronto, ON, Canada) following the manufacturer’s instructions. RNA quantity and quality were determined using a Nanodrop ND-10000V3.6.0 (NanoDrop Technologies, Wilmington, DE, USA) and Agilent 2100 Bioanalyzer (Agilent Technologies, Mississauga, ON, Canada), respectively. Samples were treated with DNase I to digest genomic DNA, and RNA was then reverse-transcribed using Superscript II (Invitrogen, Waltham, MA, USA) according to the manufacturer’s protocols. The reverse-transcription reaction was performed on 2 μg of RNA. The mRNA expression levels were estimated using quantitative PCR (Step One Plus, Applied Biosystems, Burlington, ON, Canada) with TaqMan Gene Expression Master Mix (Applied Biosystems, Burlington, ON, Canada) and TaqMan Gene Expression Assay kits, AMPK for rat Rn00576935_m1; CPT1α for rat Rn00580702_m1; ELOVL2 for rat Rn01450663_m1; FADS2 for rat Rn00580220_m1; FASN for rat Rn00569117_m1; FOXO1 for rat Rn01494868_m1; MTTP1 for rat Rn01522963_m1; PPARγ for rat Rn00440945_m1; SCD1 for rat Rn06152614_s1; SREBP1 for rat Rn01495769_m1; β-actin for rat Rn00667869_m1 (Applied Biosystems, Burlington, ON, Canada). The conditions set were as follows: 50 °C for 1 min and 95 °C for 10 min, followed by 40 cycles of 95 °C for 15 s and 60 °C for 30 s. Relative RNA expression for each gene in a sample was standardized to the endogenous housekeeping gene β-actin and calculated using the 2^−ΔΔCT^ method, with the tumor-bearing group as the comparison group.

### 4.4. Determination of Leptin and IL-4

Frozen rat livers (~50 mg) were pulverized in liquid nitrogen using a mortar and pestle and homogenized (1:10 *w:v* ratio) in an extraction buffer (20 mM Tris HCl pH 7.5; 0.5% Tween 20; 150 mM NaCl and protease inhibitors 1:100) with glass beads (0.5 mm diameter; Fast Prep ^®^-24, MP Biomedicals, Santa Ana, CA, USA) for 25 s then placed on ice. All samples were diluted to same protein concentration, 1.6 mg/mL. LuminexxMAP technology was performed using the Luminex™ 200 system (Luminex, Austin, TX, USA). Leptin and IL-4 were measured in the samples using Rat Cytokine Multi Plex Discovery Assay^®^ (Millipore Sigma, Burlington, MA, USA) according to the manufacturer’s protocol. Assay sensitivities of these markers range from 0.3–30.7 pg/mL.

### 4.5. Statistical Analysis

Data are reported as means ± SD. To identify significant differences between groups, data were evaluated using a one-way ANOVA or Student’s *t*-test. Normality of data was tested using Shapiro–Wilk’s Statistic (W > 0.05 for normally distributed data). If the data did not follow a normal distribution even if transformed, a nonparametric test was used (Kruskal–Wallis). Post hoc analysis was performed by Duncan multiple comparison tests for simple effect when interactions were present, or the Kruskal–Wallis test indicated the presence of differences. A *p*-value of < 0.05 was considered a significant difference. Data was analyzed by using SPSS Statistics for Windows, Version 28.0 (IBM SPSS Statistics for Windows, Version 28.0. Armonk, NY, USA: IBM Corp).

## 5. Conclusions

In conclusion, after chemotherapy treatment with 5-FU+CPT-11, a significant increase in hepatic fatty acid content with a depletion of EPA content was found in a model of colorectal cancer. Dietary fish oil initiated at the same time as chemotherapy reduced expression of lipogenic genes and upregulated long-chain PUFA converting genes. Dietary fish oil also upregulated genes involved in mitochondrial β-oxidation and lipid transport. Future exploration is required to assess liver capacity for fat metabolism over a longer period and after several subsequent chemotherapy cycles, as well as with the provision of dietary fish oil.

## Figures and Tables

**Figure 1 ijms-24-03547-f001:**
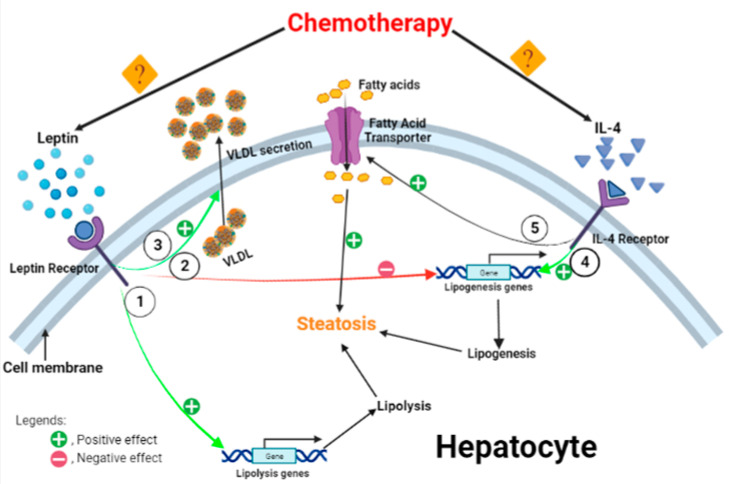
Effect of leptin and IL-4 in hepatic lipid metabolism. Leptin regulates lipid metabolism by increasing expression of genes involved in lypolysis (1), by reduction of expression of genes involved in lypolysis (2), and/or by increasing secretion of fatty acids from hepatocytes in the form of VLDL (3). Interleukin (IL)-4 regulates hepatic lipid metabolism by increasing expression of genes involved in lipogenesis (4) and/or by increasing fatty acid transport into the hepatocyte (5). The effects of chemotherapy on leptin and IL-4 are not yet known.

**Figure 2 ijms-24-03547-f002:**
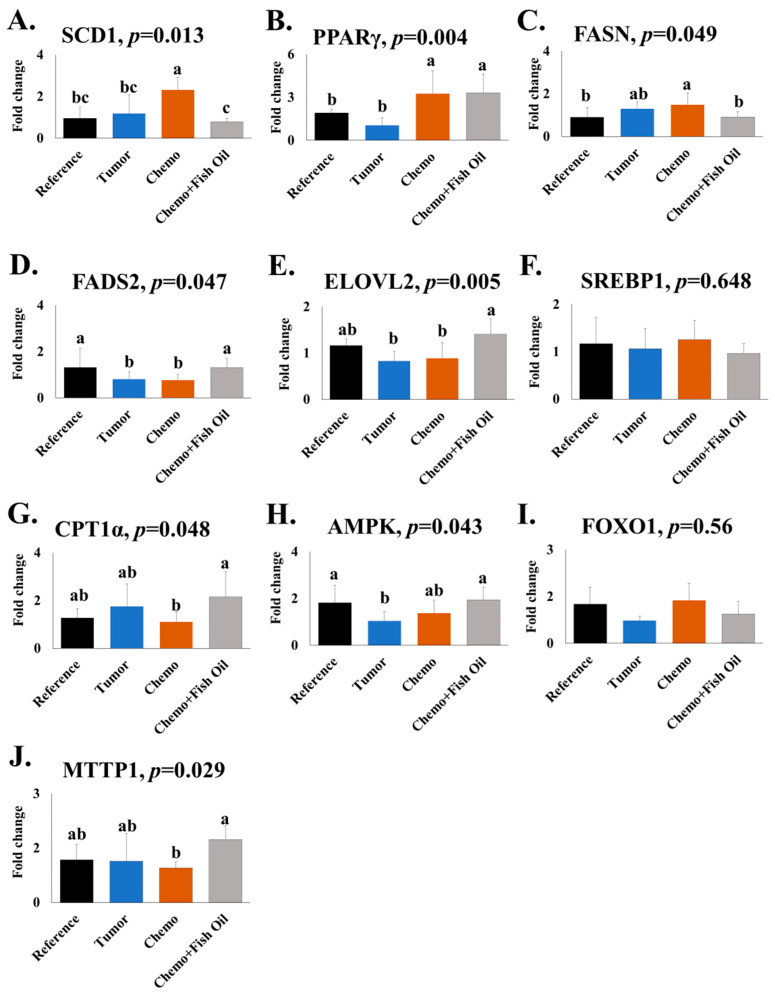
Expression of genes involved in fatty acid metabolism. Relative gene expression calculated using the 2^−ΔΔCT^ method. Values are expressed as mean ± SD. Significant differences (*p* < 0.05) were determined using one-way ANOVA. Values with differing letters are significantly different from each other (a > b > c). Groups: **Reference**, healthy rats did not undergo tumor implantation nor receive chemotherapy and consumed a control diet; **Tumor**, tumor-bearing animals did not receive chemotherapy and consumed a control diet; **Chemo**, tumor-bearing animals received chemotherapy and consumed a control diet; **Chemo** + **Fish Oil**, tumor-bearing animals received chemotherapy and consumed a fish-oil diet that started the same day as chemotherapy. Abbreviations: AMPK, activated protein kinase; CPT1α, carnitine palmitoyl transferase 1; ELOVL2, fatty acid elongases 2; FADS2, delta-6 desaturase; FASN, fatty acid synthase; FOXO1, forkhead box O1; MTTP1, triglyceride transfer protein-1; PPARγ, peroxisome proliferator-activated receptor gamma; SCD1, stearoyl-CoA desaturase; SREBP1, sterol retinol ester binding protein-1.

**Figure 3 ijms-24-03547-f003:**
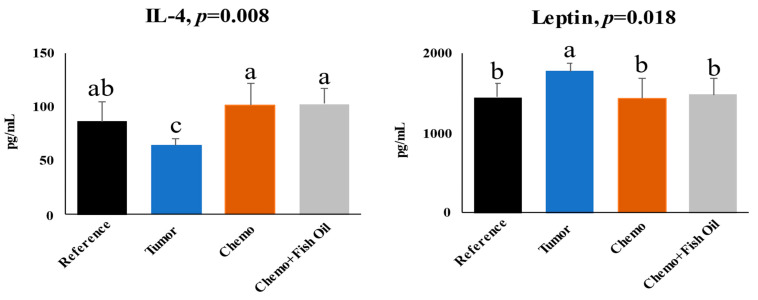
Hepatic interleukin (IL)-4 and leptin. Values are expressed as mean ± SD. Significant differences (*p* < 0.05) were determined using a one-way ANOVA. Values with differing letters are significantly different from each other (a > b > c). Groups: **Reference**, healthy rats did not undergo tumor implantation nor receive chemotherapy and consumed a control diet; **Tumor**, tumor-bearing animals did not receive chemotherapy and consumed a control diet; **Chemo**, tumor-bearing animals received chemotherapy and consumed a control diet; **Chemo** + **Fish Oil**, tumor-bearing animals received chemotherapy and consumed a fish-oil diet that started the same day as chemotherapy.

**Figure 4 ijms-24-03547-f004:**
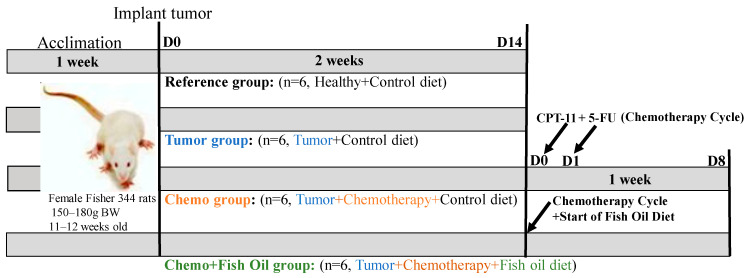
Study design. Abbreviation: Chemo, chemotherapy; CPT-11, irinotecan; D, day; 5-FU, 5-fluorouracil.

**Table 1 ijms-24-03547-t001:** (a) Hepatic TG fatty acid composition; (b) Hepatic PL fatty acid composition.

**(a)**					
Fatty Acid (%)	Reference	Tumor	Chemo	Chemo + Fish Oil	*p*-Value
C16:0	19.6 ± 0.4 ^b^	22.0 ± 1.4 ^a^	18.6 ± 1.3 ^b^	17.6 ± 1.8 ^b^	<0.001
C18:0	7.1 ± 0.1	6.9 ± 0.5	7.9 ± 0.1	7.95 ± 1.1	0.058
C18:1n-9	47.6 ± 1.7 ^a^	46.7 ± 2.31 ^a^	49.9 ± 2.9 ^a^	37.2 ± 1.2 ^b^	0.001
C18:2n-6	16.1 ± 1.0	15.0 ± 1.2	14.1 ± 2.5	13.8 ± 3.0	0.270
C18:3n-3	1.2 ± 0.4	1.0 ± 0.1	0.9 ± 0.2	0.9 ± 0.1	0.060
C20:4n-6	2.5 ± 0.7	2.5 ± 0.6	2.4 ± 0.7	2.2 ± 0.3	0.873
C20:5n-3	0.3 ± 0.1 ^bc^	0.3 ± 0.1 ^b^	0.2 ± 0.1 ^c^	6.0 ± 0.8 ^a^	0.003
C22:5n-3	0.4 ± 0.1 ^b^	0.4 ± 0.1 ^b^	0.4 ± 0.1 ^b^	3.6 ± 0.8 ^a^	0.008
C22:6n-3	1.5 ± 0.5 ^b^	1.3 ± 0.2 ^b^	1.1 ± 0.3 ^bc^	8.1 ± 1.9 ^a^	0.002
∑n-6 FA	18.8 ± 1.8	17.7 ± 1.8	16.8 ± 3.3	16.2 ± 3.4	0.248
∑n-3 FA	3.6 ± 1.2 ^b^	3.0 ± 0.5 ^bc^	2.7 ± 0.8 ^bcd^	18.8 ± 3.5 ^a^	0.001
n-6/n-3	5.2 ± 1.5 ^a^	5.8 ± 3.3 ^a^	6.2 ± 4.0 ^a^	0.9 ± 0.9 ^b^	0.002
∑SFA	27.0 ± 1.5 ^b^	29.1 ± 2.0 ^a^	26.8 ± 1.6 ^b^	25.7 ± 2.9 ^b^	0.004
∑MUFA	49.8 ± 2.6 ^a^	48.5 ± 2.6 ^a^	51.2 ± 3.2 ^a^	38.4 ± 1.3 ^b^	0.002
∑Total FA (μg/g)	2293.5 ± 909.9 ^ab^	1683.4 ± 642.7 ^b^	3243.8 ± 848.0 ^a^	2624.1 ± 654.5 ^ab^	0.024
**(b)**					
**Fatty Acid (%)**	**Reference**	**Tumor**	**Chemo**	**Chemo + Fish Oil**	***p*-Value**
C16:0	8.1 ± 0.1	8.8 ± 0.4	9.5 ± 2.3	9.4 ± 1.9	0.104
C18:0	37.1 ± 0.8 ^ab^	38.3 ± 1.1 ^a^	35.6 ± 2.2 ^b^	36.1 ± 1.2 ^b^	0.017
C18:1n-9	4.5 ± 0.5 ^b^	4.8 ± 0.5 ^b^	5.6 ± 0.5 ^a^	5.3 ± 0.5 ^ab^	0.007
C18:2n-6	8.1 ± 0.5	8.6 ± 0.4	8.7 ± 1.6	8.4 ± 1.3	0.344
C18:3n-3	0.06 ± 0.05	0.02 ± 0.03	0.03 ± 0.04	0.05 ± 0.06	0.533
C20:4n-6	29.9 ± 0.6 ^a^	26.8 ± 1.2 ^bc^	28.1 ± 2.5 ^b^	20.2 ± 0.9 ^d^	<0.001
C20:5n-3	0.5 ± 0.1 ^c^	0.7 ± 0.1 ^b^	0.5 ± 0.1 ^c^	6.7 ± 0.7 ^a^	<0.001
C22:5n-3	0.6 ± 0.1 ^b^	0.6 ± 0.1 ^b^	0.5 ± 0.3 ^bc^	1.5 ± 0.1 ^a^	0.006
C22:6n-3	9.3 ± 1.2 ^b^	9.5 ± 1.1 ^b^	9.4 ± 0.5 ^b^	11.6 ± 0.9 ^a^	<0.001
∑n-6 FA	39.0 ± 0.8 ^a^	36.5 ± 1.2 ^bc^	37.6 ± 1.4 ^b^	29.3 ± 2.1 ^d^	<0.001
∑n-3 FA	10.4 ± 1.4 ^b^	10.8 ± 1.1 ^b^	10.4 ± 0.79 ^b^	19.8 ± 1.3 ^a^	0.004
n-6/n-3	3.8 ± 0.6 ^a^	3.4 ± 0.4 ^a^	3.6 ± 0.3 ^a^	1.5 ± 0.2 ^b^	0.003
∑SFA	45.6 ± 0.7	47.2 ± 1.4	45.4 ± 1.7	45.9 ± 1.3	0.124
∑MUFA	4.8 ± 0.5 ^b^	5.2 ± 0.5 ^ab^	5.9 ± 0.6 ^a^	5.6 ± 0.5 ^a^	0.007
∑Total FA (μg/g)	27.9 ± 2.9 ^ab^	24.0 ± 2.2 ^b^	33.6 ± 4.0 ^a^	32.0 ± 6.3 ^a^	0.003

(a) Fatty acids within liver TG were determined by gas chromatography. Total amount of liver TG (μg/g) was calculated using 25 μL of C15:0 standard (0.1 μg/μL). Individual fatty acids were determined as proportion (%) of total TG; (b) Amount of fatty acids in liver PL were determined by gas chromatography. Total liver PL were calculated using 50 μg of C17:0 standard (0.1 μg/μL) to determine the μg of total fatty acids. Individual fatty acids were determined as proportionate amount (%) of total PL. Values are expressed as mean ± standard deviation. Significant differences (*p* < 0.05) were determined using a one-way ANOVA. Values with differing superscript letters are significantly different from each other (a > b > c). Groups: **Reference**, healthy rats did not undergo tumor implantation nor receive chemotherapy and consumed a control diet; **Tumor**, tumor-bearing animals did not receive chemotherapy and consumed a control diet; **Chemo**, tumor-bearing animals received chemotherapy and consumed a control diet; **Chemo** + **Fish Oil**, tumor-bearing animals received chemotherapy and consumed a fish-oil diet that started the same day as chemotherapy. Abbreviations: FA, fatty acid; MUFA, monounsaturated fatty acids; SFA, saturated fatty acids; TG, triglyceride; PL, phospholipid.

**Table 2 ijms-24-03547-t002:** Fatty acid composition of experimental diets (% of total fatty acids in the lipid source).

	Control Diet	Fish Oil Diet
Saturated fatty acids	58.7	59.9
Monounsaturated fatty acids	17.3	14.3
Polyunsaturated fatty acids	20.6	22.5
Total n-6	18.6	13.6
Total n-3	2.00	8.90
EPA	Nil	5.10
DHA	Nil	2.10
Other fatty acids	3.40	3.30

Diets were isocaloric and isonitrogenous. Fatty acid composition was measured by gas chromatography. DHA, docosahexaenoic acid; EPA, eicosatetraenoic acid.

## Data Availability

The data presented in this study are available on request from the corresponding author.

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
