# Peer review of "Increased Expression of Hepatic Stearoyl-CoA Desaturase (SCD)-1 and Depletion of Eicosapentaenoic Acid (EPA) Content following Cytotoxic Cancer Therapy Are Reversed by Dietary Fish Oil"

_ijms, 2023, doi:10.3390/ijms24043547_

Round 1

Reviewer 1 Report

The manuscript by Monirujjaman and colleges investigates whether dietary fish oil mitigates the chemo-induced liver fatty acid metabolism change, in in vivo studies conducted in rats by measuring triacylglycerol (TG), phospholipid (PL), ten lipid metabolism genes, leptin and IL-4. The in vivo experiments revealed that fish oil diet restored depletion of EPA content and overexpression of SCD-1 associated to cancer therapy, which is a very interesting result.

Abstract : define the animal gender

Introduction : line 46 - missing a space following the word “acids”

Results:

Pls correct, in line 91, palmitic acid (16:0) by palmitic acid (C16:0)

Pls, like with palmitic acid (16:0) and oleic acid (C18:1n-9) do the same for EPA (line 94) and DHA (line 95)

Table 1a & Table 1b – pls replace column title “Fish oil” by “Chemo+ Fish oil”

Legend of table 1b. – line 133 – pls delete TG, triglyceride

Fig.1 – the title of this figure is missing; pls add

Fig1 – pls include in the legend the genes

Sub-heading 2.4 – line 151 – replace IL-4 by interleukin (IL)-4

Line 153 – write “chemothewrapy+dietary fish oil” instead of “dietary fish oil”

Please explain better the sentence placed in all legends “Values with differing lower case letters indicate simple effect differences between values (a<b<c)

Discussion:

Line 170 – pls name the enzymes

Line 181 – pls write down the acronyms NAFLD, as it looks that it is the first time it appears

Line 186 – Chemo makes hepatic EPA content and also DHA content to decrease, doesn’t it? Pls revise

Line 195 – “no study has evaluated the effect of dietary n-3 PUFA in cancer animal model providing chemo treatments”. Pls revise this sentence according to publish data, as for example:

 “Fish oil mitigates myosteatosis and improves chemotherapy efficacy in a preclinical model of colon cancer”, Almasud et al, 2017

“Dietary EPA+DHA Mitigate Hepatic Toxicity and Modify the Oxylipin Profile in an Animal Model of Colorectal Cancer Treated with Chemotherapy”, Monirujjaman et al, 2022

Line 212 – pls revise the sentence “SCD1 (51) and (52) and PPAR…” for SCD1 (51, ) and (52) and PPAR…

Material and methods

pls explain better the calculation for extrapolation of human to animal dosages of the regimen CPT-11+5-FU

pls explain better the calculation for extrapolation of animal to human dosages of the fish oil diet, and its relevance

pls explain better the calculation of the administered animal dose of the fish oil through the diet. How much did in fact each animal ingested?

Table 2 – pls include the two missing column titles

Legend of table 2. –pls delete “AIN, American Institute of Nutrition”

-The timeline of the in vivo study is hard to parse from the methodology. It would be very helpful to have a diagram of the timeline for when each of drug treatments happened and when the sacrifice occurred.

Conclusions:

Line 384 - pls replace “Dietary EPA+DHA” by “Dietary fish oil”

Author Response

Response to Reviewer 1 Comments

The manuscript by Monirujjaman and colleges investigates whether dietary fish oil mitigates the chemo-induced liver fatty acid metabolism change, in in vivo studies conducted in rats by measuring triacylglycerol (TG), phospholipid (PL), ten lipid metabolism genes, leptin and IL-4. The in vivo experiments revealed that fish oil diet restored depletion of EPA content and overexpression of SCD-1 associated to cancer therapy, which is a very interesting result.

Point 1: Abstract : define the animal gender

Response 1: Defined sex.

Point 2: Introduction : line 46 - missing a space following the word “acids”

Response 2: Corrected.

Results:

Point 3: Pls correct, in line 91, palmitic acid (16:0) by palmitic acid (C16:0)

Response 3: Corrected to C16:0 and subsequent ones.

Point 4: Pls, like with palmitic acid (16:0) and oleic acid (C18:1n-9) do the same for EPA (line 94) and DHA (line 95)

Response 4: Included C20:5n-3 for EPA and C22:6n-3 for DHA.

Point 5: Table 1a & Table 1b – pls replace column title “Fish oil” by “Chemo+ Fish oil”

Response 5: Updated accordingly.

Point 6: Legend of table 1b. – line 133 – pls delete TG, triglyceride

Response 6: Deleted.

Point 7: Fig.1 – the title of this figure is missing; pls add

Response 7: Included title.

Point 8: Fig1 – pls include in the legend the genes

Response 8: Added.

Point 9: Sub-heading 2.4 – line 151 – replace IL-4 by interleukin (IL)-4

Response 9: Replaced.

Point 10: Line 153 – write “chemotherapy+dietary fish oil” instead of “dietary fish oil”

Response 10: Replaced with chemotherapy+dietary fish oil

Point 11: Please explain better the sentence placed in all legends “Values with differing lower case letters indicate simple effect differences between values (a<b<c)

Response 11: We have rewritten the sentence as “Values with differing superscript letters are significantly different from each other (a>b>c)” for tables and “Values with differing letters are significantly different from each other (a>b>c)” for figures.

Discussion:

Point 12: Line 170 – pls name the enzymes

Response 12: Included names.

Point 13: Line 181 – pls write down the acronyms NAFLD, as it looks that it is the first time it appears

Response 13: Defined NAFLD.

Point 14: Line 186 – Chemo makes hepatic EPA content and also DHA content to decrease, doesn’t it? Pls revise

Response 14: In the present study we observed that chemotherapy significantly reduced EPA content in the liver, compared to tumor bearing animals; while, the reduction in DHA content in response to chemotherapy was insignificant. 

Point 15: Line 195 – “no study has evaluated the effect of dietary n-3 PUFA in cancer animal model providing chemo treatments”. Pls revise this sentence according to publish data, as for example:

 “Fish oil mitigates myosteatosis and improves chemotherapy efficacy in a preclinical model of colon cancer”, Almasud et al, 2017

“Dietary EPA+DHA Mitigate Hepatic Toxicity and Modify the Oxylipin Profile in an Animal Model of Colorectal Cancer Treated with Chemotherapy”, Monirujjaman et al, 2022

Response 15: We have rewritten the sentence to read that “However, to date no study has determined the effect of dietary fish oil on specific hepatic perturbations in fatty acids following chemotherapy, and what changes in genes may be underlying these changes.

Point 16: Line 212 – pls revise the sentence “SCD1 (51) and (52) and PPAR…” for SCD1 (51, ) and (52) and PPAR…

Response 16: We have rewritten the sentence as “expression of SCD1 [51, 52] and PPARγ [53] have been observed”…..

Material and methods

Point 17: Pls explain better the calculation for extrapolation of human to animal dosages of the regimen CPT-11+5-FU

Response 17: Irinotecan plus 5-fluorouracil [CPT-11 plus 5-FU] therapy is the most commonly applied chemotherapy regimen. The combination of this therapy is also known as FOLFORI.   During the development of this model, the balance between drug effectiveness to the tumor and the toxicities to the host were carefully explored to ensure maximal tumor efficacy with limited toxicity (outlined in reference 77; Cao et al., 2000).  Since establishing this model, we have been applying this model to the understanding of chemotherapy induced changes in metabolism and nutritional interventions that modify these effects (references 78 and 80, for example). We have outlined this in our manuscript in lines 346-348 ”This experimental model carefully recapitulates therapy for colorectal cancer in humans with respect to doses and toxicity of a combined regimen of CPT-11 plus 5-FU [77, 78, 80]”.

In our model 50mg/kg CPT-11 and 5-FU equates to approximately 7.5 mg dose of each drug to our animals weighing ~150 g. 

Point 18: pls explain better the calculation for extrapolation of animal to human dosages of the fish oil diet, and its relevance

Response 18: In a prior clinical trial, patients receiving first-line chemotherapy for advanced non-small cell lung cancer lost skeletal muscle mass and gained intermuscular fat, while patients who supplemented with EPA and DHA (2.1 g/day) beginning on the first day of chemotherapy enhanced the tumor response while resulting in a maintenance or gain in skeletal muscle mass and reduction in intermuscular fat over the same time period (ref 75). We use this same effective dose in our study here, with the diets provided differing only in EPA and DHA content which was incorporated into a semi-purified diet otherwise resembling westernized human diets with respect to macronutrient content. We have updated subsection 2.1. General Findings pls see lines 95-96, we have also included a paragraph in discussion section to address this, pls find lines 286-291. We have also rewritten subsection 4.1. Animal Handling, Diet and Experimental Design for better understanding pls find lines 317-337.

Point 19: pls explain better the calculation of the administered animal dose of the fish oil through the diet. How much did in fact each animal ingested?

Response 19:  Fish oil was added to provide EPA+DHA at a dose similar to what is observed in human clinical trials. Rats consumed approximately 0.5mg/kg BW per day. Most clinical trials provide 2-4 g of fish oil per day which equates to approximately 0.3-0.6mg/kg per day and aligns with the dose provided in this study.

We have rewritten subsection 4.1. Animal Handling, Diet and Experimental Design for better understanding pls find lines 317-337. We have updated subsection 2.1. General Findings pls see lines 95-96.

Point 20: Table 2 – pls include the two missing column titles

Response 20: Added

Point 21: Legend of table 2. –pls delete “AIN, American Institute of Nutrition”

Response 21: Deleted

Point 22: -The timeline of the in vivo study is hard to parse from the methodology. It would be very helpful to have a diagram of the timeline for when each of drug treatments happened and when the sacrifice occurred.

Response 22: We have added study design figure as Figure 4.

Conclusions:

Point 23: Line 384 - pls replace “Dietary EPA+DHA” by “Dietary fish oil”

Response 23: Replaced with dietary fish oil.

Reviewer 2 Report

  The manuscript concerns an important aspect related to patients' diet during chemotherapy. I have a few minor comments that should be made to the manuscript:

-complete the literature after the sentence „CRC most commonly metastasizes to the liver which is treated surgically in 33 ~ 15-20% for increased long-term survival” (line 33-34)

-sentence „Patients with advanced cancer (of various tumor types) exhibit a deficit of 30-50% of plasma PL and constituent fatty acids compared with healthy subjects [16]” is unclear. Is regards the pateints with adavnced cancer or patients with advanced cancer after chemiotherpay ? (line 56-57)

- the introduction asks for a diagram illustrating the connections of all described factors such as leptins, interleukins and others in fatty acid metabolism

- the manuscript lacks a clear indication of the element of novelty - whether such research is being carried out for the first time, or maybe it has already been carried out, or what similar studies have been carried out in this direction

Author Response

Response to Reviewer 2 Comments

The manuscript concerns an important aspect related to patients' diet during chemotherapy. I have a few minor comments that should be made to the manuscript:

Point 1: -complete the literature after the sentence „CRC most commonly metastasizes to the liver which is treated surgically in 33 ~ 15-20% for increased long-term survival” (line 33-34)

Response 1: We have added ref [2] for this information.

Point 2: -sentence „Patients with advanced cancer (of various tumor types) exhibit a deficit of 30-50% of plasma PL and constituent fatty acids compared with healthy subjects [16]” is unclear. Is regards the pateints with adavnced cancer or patients with advanced cancer after chemiotherpay ? (line 56-57)

Response 2:  We have rewritten the sentence as “Patients with advanced cancer (of various tumor types) exhibit a deficit of 30-50% of plasma PL and constituent fatty acids compared with healthy subjects [17], which may be exacerbated by chemotherapy treatment”. Please see lines 56-58.

Point 3: - the introduction asks for a diagram illustrating the connections of all described factors such as leptins, interleukins and others in fatty acid metabolism

Response 3: We have added figure 1 showing effect of leptin and IL-4 on hepatic lipid metabolism.

Point 4: - the manuscript lacks a clear indication of the element of novelty - whether such research is being carried out for the first time, or maybe it has already been carried out, or what similar studies have been carried out in this direction

Response 4: To date no study has determined the effect of dietary fish oil on specific hepatic perturbations in fatty acids following chemotherapy, nor evaluated how genes related to fat metabolism may be involved. We indicated this in our study. Please find in lines 191-193.